# Bacteria–Fungi Interactions in Multiple Sclerosis

**DOI:** 10.3390/microorganisms12050872

**Published:** 2024-04-26

**Authors:** Miriam Gorostidi-Aicua, Iraia Reparaz, Ane Otaegui-Chivite, Koldo García, Leire Romarate, Amaya Álvarez de Arcaya, Idoia Mendiburu, Maialen Arruti, Tamara Castillo-Triviño, Laura Moles, David Otaegui

**Affiliations:** 1Biogipuzkoa Health Research Institute, Neuroimmunology Group, 20014 San Sebastián, Spain; miriam.gorostidiaicua@bio-gipuzkoa.eus (M.G.-A.); iraia.reparazbonilla@bio-gipuzkoa.eus (I.R.); ane.otaeguichivite@bio-gipuzkoa.eus (A.O.-C.); koldo.garciaetxebarria@bio-gipuzkoa.eus (K.G.); leire.romarategarcia@bio-gipuzkoa.eus (L.R.); idoia.mendiburuarrieta@osakidetza.eus (I.M.); maialen.arrutigonzalez@osakidetza.eus (M.A.); tamara.castillotrivino@osakidetza.eus (T.C.-T.); 2Center for Biomedical Research Network in Neurodegenerative Diseases (CIBER-CIBERNED-ISCIII), 28029 Madrid, Spain; 3Neurology Department, Osakidetza Basque Health Service, Hospital Universitario Araba, 01009 Vitoria-Gasteiz, Spain; amayaelena.alvarezdearcayaesquide@osakidetza.eus; 4Neurology Department, Osakidetza Basque Health Service, Hospital Universitario Donostia, 20014 San Sebastián, Spain

**Keywords:** mycobiome, bacteriome, multiple sclerosis, ion torrent sequencing, NGS

## Abstract

Multiple sclerosis (MS) arises from a complex interplay between host genetic factors and environmental components, with the gut microbiota emerging as a key area of investigation. In the current study, we used ion torrent sequencing to delve into the bacteriome (bacterial microbiota) and mycobiome (fungal microbiota) of people with MS (pwMS), and compared them to healthy controls (HC). Through principal coordinate, diversity, and abundance analyses, as well as clustering and cross-kingdom microbial correlation assessments, we uncovered significant differences in the microbial profiles between pwMS and HC. Elevated levels of the fungus *Torulaspora* and the bacterial family *Enterobacteriaceae* were observed in pwMS, whereas beneficial bacterial taxa, such as *Prevotelladaceae* and *Dialister*, were reduced. Notably, clustering analysis revealed overlapping patterns in the bacteriome and mycobiome data for 74% of the participants, with weakened cross-kingdom interactions evident in the altered microbiota of pwMS. Our findings highlight the dysbiosis of both bacterial and fungal microbiota in MS, characterized by shifts in biodiversity and composition. Furthermore, the distinct disease-associated pattern of fungi–bacteria interactions suggests that fungi, in addition to bacteria, contribute to the pathogenesis of MS. Overall, our study sheds light on the intricate microbial dynamics underlying MS, paving the way for further investigation into the potential therapeutic targeting of the gut microbiota in MS management.

## 1. Introduction

Multiple sclerosis (MS) stands as an immune-mediated demyelinating disease of the central nervous system, that is the leading cause of non-traumatic neurological disability in young adults aged 20–40 years [1,2]. Due to its complex pathophysiology, MS has a heterogeneous clinical presentation and course, with distinct clinical forms. Relapsing–remitting MS (RR MS) accounts for approximately 85% of cases and is characterized by episodes of neurological disability symptoms lasting at least 24 h (relapses), followed by periods of complete or incomplete recovery (remission) [3,4].

The wide phenotypic diversity, coupled with unpredictable clinical relapses and remissions, make the treatment and prognosis of MS quite challenging. Despite progress in understanding the pathophysiology of the disease in recent decades, many mysteries remain. In recent years, the study of the gut microbiota has been incorporated in an attempt to unravel the etiology and pathophysiology of the disease.

The gut microbiota is a complex ecosystem consisting of thousands of microorganisms that are critical for human health, including the development and regulation of the immune system. While much of the research on the microbiota focuses on sequencing the 16S rRNA gene, which is specific to prokaryotes, this focus often overlooks other minority microorganisms that play a key role in maintaining the microbiota structure and interacting with the host to support various functions. Among these microorganisms, the population of yeasts and fungi, collectively known as the mycobiota, is emerging as a major contributor to the complex dynamics of gut health.

Previous studies have characterized human mycobiota as dynamic and variable between individuals, with their presence detected in the gut of approximately 70% of healthy adults [5]. In particular, research has identified at least 75 genera and 267 species within the human gut mycobiota [6,7], with the phyla Ascomycota and Basidiomycota dominating and certain genera such as *Candida*, *Saccharomyces*, *Cladosporidium*, *Malassezia*, and *Aspergillus* emerging as predominant [6,8,9,10]. While the mycobiota typically represents between 0.1% and 1% of the total microbiota [11], some studies suggest that its representation could be even higher [10]. Despite its relatively low abundance, the ability of the mycobiota to modulate the host immune system has significant implications for human health [12], with associations identified between mycobiota dysbiosis and several conditions, including chronic inflammation, cystic fibrosis, inflammatory bowel syndrome, obesity, autism, schizophrenia, and Alzheimer’s disease [13,14,15,16,17].

Although research into the role of the mycobiota in MS is still in its infancy, several hypotheses suggest that the mycobiota may be altered in individuals with the disease. The profile of the gut mycobiome shows variability influenced by factors such as age, gender, diet, and medication [15,18,19]. In addition, certain biomarkers of MS, such as the HLA-DRB1*15 allele or the presence of inflammatory markers like chitotriosidase and calprotectine in cerebrospinal fluid have been associated with fungal infections [20,21,22].

In recent years, there has been an increasing focus on interaction between bacterial species; however, fungi, which occupy the same niche as bacteria, also play an important role in influencing the composition and function of bacterial communities. These cross-kingdom interactions give rise to complex and dynamic networks of microorganisms that secrete molecules capable of either promoting or inhibiting the growth of neighboring populations, potentially leading to the production of antifungal or antibacterial compounds [23,24]. Indeed, several studies have identified specific bacterial–fungal interactions associated with the development and severity of disease [25]. However, most published studies have separately analyzed either bacterial or fungal communities. Thus, the simultaneous analysis of both kingdoms from the same sample or patient remains relatively limited, despite the potential importance of deciphering the mechanisms operating in a competitive polymicrobial environment for understanding microbial dysbiosis in disease development and progression [26,27,28]. Furthermore, bacterial–fungal interactions have been shown to influence the immune response, with co-infections involving both pathogens inducing different responses from infections with either pathogen alone [23].

The simultaneous study of bacterial and fungal populations from a single sample is challenging as it requires the mechanical, chemical, and/or enzymatic cell disruption of both types of microorganisms [29]. In addition, primer selection for amplification and sequencing is not standardized, particularly for fungal communities, and there is no gold standard for taxonomical identification in the subsequent data processing [30,31].

Although initial steps have been taken to study the mycobiota in the context of MS, the relationship between mycobiota and the disease remains to be fully elucidated. Investigating bacterial–fungal interactions could provide valuable insights into disease understanding and progression. In this study, we aim to profile the gut microbiota and mycobiota of RR MS patients (*n* = 62) and healthy controls (*n* = 36) to assess changes in fungal population alterations in pwMS and further explore the crosstalk between fungi and bacteria in the disease. Our aim is to gain a broader understanding of the functionality of the microbiome as a whole in the context of MS.

## 2. Materials and Methods

### 2.1. Ethics

The research protocol was approved by the local ethics committee with the reference PI2020070 on the 14 July 2021. Written informed consent was attained from all participants involved in the study.

### 2.2. Participants and Sample Collection

The study included 98 stool samples, 62 from individuals with MS and 36 from healthy controls (HC) recruited from the Hospital Universitario Donostia (Donostia-San Sebastián, Spain).

To be eligible for enrollment, participants must meet specific criteria: they must not have gastrointestinal or chronic infectious diseases, have received steroid treatment within the past month or undergone chemotherapy or antibiotic therapy within the last three months. In addition, participants must not be pregnant or within six months post-delivery.

For HC, in addition to not being related to any pwMS, individuals must not have autoimmune, gastrointestinal, or chronic infectious diseases. These strict eligibility criteria ensure a homogeneous study population and minimize confounding factors that could influence the results.

Relevant clinical and demographic data, such as age, sex, MS type, Expanded Disability Status Scale (EDSS), disease evolution, or disease-modifying treatment (DMT) were recorded for each participant (Table 1).

All participants were provided with a comprehensive kit containing all the necessary materials for hygienic fecal sampling in the comfort of their own home. The kit included a stool collector designed to adhere to the toilet, tubes to collect stool samples, a safety bag, and hydrated cold packs to maintain the required temperature. An isothermal bag was also provided for safe transport of the samples.

Upon collection, participants deposited their stool sample into the provided tube, which was immediately frozen at −20 °C. The samples were then transported to the hospital, using the cold packs included in the kit. On arrival at our center, the samples were thoroughly checked for suitability before being stored at −80 °C until further analysis.

### 2.3. DNA Extraction

Fecal samples were thawed on ice and diluted in 1X Dulbecco’s phosphate-buffered saline (DPBS) (Gibco BRL, Gaithersburg, MD, USA). DNA was then extracted trough the mechanical and enzymatic lysis of the cells and the extraction kit QIAamp DNA Stool Mini Kit (Qiagen, Hilden, Germany). Mechanical disruption was performed by bead-beating with 0.1 mm diameter zirconia/silica beads three times (Sigma, St. Louis, MO, USA) using a Tissue Lyser (Qiagen, Germany). Enzymatic lysis of fungal cells was performed using the enzyme zymolyase (MP Biomedicals, LLC, Illkirch-Graffenstaden, France) at a concentration of 0.1 mg/mL. The enzyme lysozyme (Sigma, St. Louis, MO, USA) at a concentration of 10 mg/mL was used for the bacterial lysis. The obtained DNA was subjected to quality and quantity controls. A negative control was extracted in parallel with fecal samples in order to evaluate the influence of the reactants and possible contamination background of the samples. This control was performed in an empty sterile tube (similar to ones containing the feces) without adding any biological sample.

### 2.4. DNA Amplification and Sequencing

Amplification of the fungal ITS1 and ITS2 intergenic regions was carried out using the following primers: ITS1 30F (5′-GTCCCTGCCCTTTGTACACA-3′) and ITS1 217R (5′-TTTCGCTGCGTTCTTCATCG-3′); and ITS86F (5′-GTGAATCATCGAATCTTTGAA-3′) and ITS4R (5′-TCCTCCGCTTATTGATATGC-3′), respectively, and the enzyme AmpliTaq Gold™ DNA Polymerase (Applied Biosystems, Foster City, CA, USA). The thermal cycling conditions were 95 °C for 10 min then 95 °C for 30 s, 55 °C for 30 s, and 72 °C for 1 min for 35 cycles. The PCR products obtained from each fecal sample were pooled and sequenced on ion torrent PGM equipment (Life Technologies, Waltham, MA, USA) using a 318 chip.

Hypervariable regions V2, V4, and V8; and V3, V6–7, and V9 of the bacterial 16S rRNA gene were amplified in separate tubes and submitted to quality controls. Between 300 and 400 ng of DNA of each sample was used to perform the amplification and sequencing. The bacterial microbiome study was performed on ion torrent PGM equipment (Life Technologies, Waltham, MA, USA) using a 318 chip.

### 2.5. Data Processing

#### 2.5.1. ITS

QIIME2 microbiome bioinformatics platform was used for ITS analysis. The pipeline performed in QIIME2 included data import, quality filter and denoising (using cutadapt for removing primers), ASV identification (by dada2 *denoise-pyro*), diversity analysis (reads were rarefied at 10,000), and taxonomic classification using the UNITE database (using the *vsearch* feature classifier). Due to the lack of available pipelines for the analysis of ion torrent sequences (partly on account of the increased use of Illumina), the best way to perform this analysis has been discussed several times by the community in the forum that QIIME2 offers. The above-explained specific steps that were carried out in our workflow were decided based on the discussions from the forum [32,33,34,35].

#### 2.5.2. 16S

For 16S analysis, after sequencing, the metagenomic workflow of the ion reporter software (https://ionreporter.thermofisher.com/ir/, accessed on 31 Jan 2022) was used in order to associate the obtained reads with bacterial taxonomies. Here, primers used from the ion torrent kit were trimmed and the reads truncated in 150. The taxonomic classification was performed using the Greengenes and MicroSEQ databases with a 97% of percentage identity for the genus level. The consensus output files were imported to R (version 4.1.0) for obtaining the relative abundance tables after merging all the sample results.

### 2.6. Statistics

The overall composition of the collected stool samples was analyzed in excel and R environment. The microbiome and mycobiome of 10 most frequent and abundant taxonomic classifications were graphed at the family–genus level and grouped by sample type (MS vs. HC). A multi-factor ANOVA and Kruskal–Wallis tests were used to compare the microbial taxon abundance within parametric and non-parametric data, respectively.

Concurrently to the descriptive analysis, the LEfSe algorithm (LDA Effect Size) from the online Galaxy Huttenhower platform was used to get the cladograms figures, where the significantly different microbial taxa are presented.

The Shannon diversity index (SDI) was calculated using the R “vegan” package. All plots were made with the “ggplot2” package, and all *p*-values were adjusted with a false discovery rate after applying the Wilcoxon test, when the variables result in non-normality by the shapiro.test function. All analyses were carried out in Rstudio version 4.1.2.

#### 2.6.1. Pearson Correlation

The relative abundances of bacteria family–genera were correlated with fungi family–genera using both the MS and control samples, with no discrimination, using the cor function from the “corrplot” package. The correlation tables and graphics were created through cor.mtest and corrplot functions, respectively, obtaining the correlation values and the *p*-values for each. Individual correlations were performed using the R “visreg” package.

#### 2.6.2. Clustering and Enterotype Identification

Before performing clustering in our data, the Hopkins function from R was used to see whether or not it made sense to pool our data in groups. In the datasets where H was different to zero, the K-means function was performed using 2 centroids. 

In order to choose the number of clusters and = analyze the quality of the resulting clusters, the strategy from Koren et al. was followed [36], maximizing the prediction strength (PS) [37] and the silhouette index (SI) [38]. We applied a criterion of 0.8 for PS for enough strength for clustering, which means that 80% of the data points entered the cluster and the rest were outliers. In the work published by Koren et al., moderate clustering was used when the SI was 0.5, as described by Wu et al., and strong clustering when the SI was >0.75 [36,39]; however, this criterion tended to not support the results obtained for PS in our data, so, even if the results of the SI were taken into account, the decision making for enterotype identification was more strongly based on the PS criterion.

## 3. Results

### 3.1. Fungal Microbiota (Mycobiota) of MS Patients

The fungal microbiome was studied in a cohort of 62 MS patients (MS) and 36 healthy controls (HC); however, we only detected some mycobiome in 34 MS patients (56%) and 28 HCs (78%). The reduction in the mycobiota was particularly pronounced in participants with a higher bacterial diversity (Shannon diversity index (SDI)) and microbiota density (µg of DNA/g of feces). Gender and age had a significant effect on the presence of mycobiota in MS patients, being lower in men and older patients (>50 years). The diversity (SDI) and richness (number of family/genus groups per sample) parameters were calculated with the fungal data, finding no statistical differences (0.43 vs. 0.23, and 11.7 vs. 9.9 in MS and HC, respectively). The heterogeneity between samples was high, but in general, 39 families/genera were exclusively in pwMS, 11 of which were only present in untreated patients; and 21 families/genera could only be detected in the mycobiota of HCs.

The number of reads in the samples had a mean of 220,844 reads, and a total of 1 to 20 family/genus (10.7 mean family/genus) were detected for each of them. While no DNA was detected in the negative control, a total of 5000 reads and 3 family/genus were detected after amplification and sequencing. Considering these data and the different profile of microorganisms obtained in the control sample, its influence on the results was considered insignificant.

The phyla Basidiomycota and especially Ascomycota dominated the samples. No differences were found at this taxonomic level, but a significant change could be observed in the Basidiomycota/Ascomycota ratio between the patients and controls (Figure 1A). Noteworthy is the high number of sequences without a taxonomic assignment, which represented a median of 6.1% in the patients and 2.0% in the HCs. Differences at the class level were limited to Dothideomycetes (*p*-value = 0.009), mainly caused by the order Pleosporales (*p*-value = 0.043) that are more abundant in the gut of pwMS. The genus *Torulospora* (*p*-value = 0.019) was also significantly more abundant in the MS mycobiota (Figure 1B,C). Only 4 of the 10 dominant genera were shared between the HCs and MS patients (*Saccharomyces*, *Candida*, *Debaryomyces*, and *Torulaspora*). The fungal profile of the dominant genera is represented in Figure 1D.

DMTs affected both fungal richness (10.7 groups) and diversity (SDI = 0.34) compared to naïve patients (10.6 groups and SDI = 0.64), even though they did not reach the levels of the HCs (9.7 groups and SDI = 0.21). The Basidiomycota/Ascomycota ratio was also modified by DMTs, reversing the trend observed in naïve MS. Other variables that influence the mycobiota composition measured by the fungal SDI and richness, and Basidiomycota/Ascomycota ratio were age, sex, EDSS, years of disease evolution, and disease prognosis (Figure 2).

### 3.2. Bacterial Microbiota of MS Patients

The characterization of the bacterial microbiota was carried out on the same samples as the fungal microbiome. No differences were found between the samples in terms of diversity (SDI) or richness parameters. However, opposite trends to the fungal population were found, i.e., a lower diversity (2.76 and 2.81 in MS and HC, respectively) and richness (58 and 58.5 bacterial groups in MS and HC, respectively) in the MS samples. Heterogeneity between the samples was lower than that found in the fungal population; however, 18 bacterial families/genera were identified exclusively in the MS samples and 70 in the HC samples.

The number of reads in the samples ranged from 6462 to 11,4262 (mean 41,189.4), and a total of 24–111 families/genera (mean 59.5 family/genus) were detected for each of them. While no DNA was detected in the negative control, a total of 424 reads and 9 families/genera were detected after amplification and sequencing. Taking these data and the different profile of microorganisms obtained in the control sample into consideration, its influence on the results was considered negligible.

No significant differences were found in the Firmicutes/Bacteroidetes ratio in MS and HCs (Figure 3A). Differences at the phyla level were limited to the phylum Actinobacteria (*p*-value = 0.018), mainly caused by the genus *Collinsella* (*p*-value = 0.017) which is more abundant in the gut of pwMS. Regarding the lower taxonomic levels, the predominance of the order Enterobacteriales (*p*-value = 0.019) and the family *Enterobacteriaceae* (*p*-value = 0.019) stood out in patients, whereas the class BetaProteobacteria (*p*-value = 0.004), order Burkhoderiales (*p*-value = 0.004), family *Prevotelladaceae* (*p*-value = 0.023), and genus *Dialister* (*p*-value = 0.043) were prominent in the gut of HCs (Figure 3B,C). The dominant genera in the samples of the HC and MS samples were almost the same (9 out 10 genera were shared between the groups) (Figure 3D).

DMTs increased diversity (SDI = 2.77) and bacterial richness (17 groups) in the gut of treated patients compared to those that were naïve (SDI = 2.69 and 57.8 groups), reaching levels similar to those observed in the controls. The Firmicutes/Bacteroidetes ratio was also modified by DMTs, reversing the trend observed in naïve MS. Other variables that influenced the bacterial composition measured by the SDI and richness, and the Firmicutes/Bacteroidetes ratio were age, sex, EDSS, years of disease evolution, and disease prognosis (Figure 4).

### 3.3. Clustering Results

The samples were subjected to clustering to identify common fungal patterns in the mycobiota of the patients and controls. When the analysis was applied to all the samples, the results showed the grouping into two clusters. The fungal groups with the greatest weight in the composition of the microbiota were the genera *Coniochaeta*, *Saccharomyces*, and *Candida* (Figure 5A). The majority of the samples were clustered as mycotype 2 (79% and 76% of HC and MS patient samples, respectively). The majority of family/genus groups differed between mycotypes, with differences observed in dominant classes such as Saccharomycetes and genera such as *Candida*. The diversity also differed between the samples grouped into each mycotype (Figure 5B). The distribution of mycotypes in the study population was analyzed considering clinical and demographic variables, and no statistical differences were found (Figure 5C). In order to identify common fungal patterns associated with disease progression, clustering analysis was repeated on patient samples only. The sample distribution was then mainly directed by the genera *Saccharomyces*, *Candida*, and *Torulaspora* (Figure 5D). The differences in dominant genera such as *Saccharomyces* and *Torulaspora* were observed (Figure 5E). Clinical and demographic data did not significantly affect the distribution of MS mycotypes in the patient samples (Figure 5F).

A similar analysis was carried out with the bacteriome. The samples were clustered in two enterotypes. The bacterial groups with the greatest weight in the composition of the microbiota were the family *Enterobacteriaceae* and the genera *Prevotella*, *Dialister*, *Bacteroides*, and *Bilophila* (Figure 6A). The majority of the samples (94.1% and 82.3% of the HC and MS patients, respectively) were clustered as enterotype 2. The dominant bacterial family/genus was the same in both enterotypes, so the clusters were separated by differences in diversity and richness, and by the different abundance of some major taxa (Figure 6B). The distribution of enterotypes in the study population was analyzed considering clinical and demographic variables. Curiously, enterotype 1 was found to be the most prevalent in the samples of patients with a higher EDSS or worse prognosis (Figure 6C).

The clustering analysis was repeated on patient samples in an attempt to identify common bacterial patterns associated with disease progression. The samples were also grouped into two clusters that were driven by the same bacterial groups as the enterotypes (Figure 6D). The majority of the samples (79%) were grouped as MS enterotype 1. The clusters were mainly separated by differences in diversity and the abundance of some major taxa (Figure 6E). Once more, the samples from patients with a higher EDSS and worse prognosis were significantly clustered together. (Figure 6F).

It is noteworthy that 74% of the samples coincide in the same cluster in the analysis carried out with the bacteriome and mycobiome data.

### 3.4. Pearson Correlation Results

The correlations between the gut mycobiome and the microbiome were studied. First, the balance between bacterial and fungal diversity in the gut was determined by the fungal-to-bacterial diversity ratio. This ratio was increased in the MS samples (0.17) and particularly in naïve MS (0.22) compared to the controls (0.09) (Figure 7A). The Spearman correlation analysis revealed different fungal–bacterial interaction in pwMS and HC at the phyla level. Significant fungal–bacterial correlations were limited to the phyla Ascomycota and Firmicutes positive correlation and the negative one between Ascomycota and Proteobacteria in HC (Figure 7B). These correlations were dispelled or inverted (in the case of Proteobacteria) in the microbiota of pwMS (Figure 7C).

Regarding the lower taxonomic level, more frequent and intense correlations were observed in the HCs than in MS (Figure 8A,B). The significant positive correlation between the bacterial families *Bacteroidaceae*, *Pasteurellaceae*, *Streptococcaceae*, and *Synergistaceae*, and the fungal genera *Malassezia*, *Meyerozima*, *Debaryomyces*, and *Candida*, respectively; the families *Pasteurellaceae* and *Saccharomycetales incertae sedis* and the bacterial genus *Parasutterella* and the fungal genus *Peniophora* were dispelled or inverted in MS microbiota (Figure 8C,D(1,6,21,22,7,19)). The negative correlation between the genera *Ruminococcus* and *Saccharomyces* in the HCs were abolished in MS microbiota. The significant positive correlation between the bacterial family *Bacteroidaceae* and the fungal family *Saccharomycetales incertae sedis*, the bacterial family *Streptococcaceae* and the fungal genera *Peniophora* and the bacterial genera *Parasutterella* and fungal family *Saccharomycetales incertae sedis* present in the gut of pwMS were abolished in the microbiota of the HCs (Figure 8C,D(3,23,15)). The bacterial genus *Ruminococcus* and the fungal genus *Saccharomyces* follow a negative correlation in the HCs and are eliminated in MS (Figure 8C,D(11)). These results suggest a shift toward a loss of complexity in bacterial–fungal interactions in MS patients.

## 4. Discussion

In this study, we investigated the gut mycobiome and bacteriome of individuals with relapsing–remitting multiple sclerosis (RR MS) during remission, and compared it to that of the healthy controls. We assessed the influence of various clinical and demographic factors on the composition of both bacterial and fungal microbiota. In addition, we explored the links between the fungal and bacterial microbiomes to elucidate the interplay between these kingdoms in the context of MS. Finally, using our dataset, we applied bioinformatic tools to discern the microbial profile of this disease and its progression.

Overall, the mycobiota in MS appeared less abundant but more diverse, enriched in fungal groups, and exhibited greater heterogeneity compared to the HC, particularly evident in treatment-naïve patients. These findings suggest that the characteristic aspects of MS may favor fungal colonization at the expense of bacterial communities. In particular, the MS mycobiota showed an enrichment of fungal taxa not identified in updated databases. Both MS and HC gut mycobiota were predominantly composed of *Saccharomyces* and *Candida*. However, *Fusarium* and *Coniochaeta* were prevalent in the gut of HCs, whereas *Torulaspora*, *Debaryomyces*, and *Cladosporidium* dominated in MS patients. While the differences at higher taxonomic levels were minimal, the Basidiomycota/Ascomycota ratio was significantly higher in MS patients. Additionally, the fungal/bacterial diversity ratio was increased in MS, particularly in treatment-naïve individuals. Recent studies have reported increased ITS/16S and Basidiomycota/Ascomycota ratios in the MS population, along with an enrichment of *Candida* and *Epicoccus* and depletion of *Saccharomyces* [40]. Although our results reflect trends in the *Candida* and *Saccharomyces* genera, they did not reach statistical significance, despite a larger cohort. Moreover, Shah et al. described an MS mycobiota profile enriched in *Saccharomyces* and *Aspergillus*, while Gargano et al. identified *Saccharomyces* and *Candida* enrichment. However, these studies used ITS1 amplicon sequencing [41,42], which may have contributed to these discrepancies. Previous research has demonstrated that ITS2 sequencing allows for an improved taxonomic identification in the gut [43]. In our study, we used a combination of ITS1 and ITS2 sequencing to provide a comprehensive analysis.

Characterization of the gut mycobiota in other inflammatory diseases such as inflammatory bowel disease (IBD) has also revealed a higher Basidiomycota/Ascomycota ratio in patients, particularly during flares [17], suggesting their potential involvement in the inflammatory process. Additionally, studies have reported mycobyota enrichment in *Candida* and depletion in *Saccharomyces*, along with an increased ITS/16S ratio in IBD. The precise role of these parameters in inflammation requires further investigation.

The role of the common yeast *Saccharomyces cerevisiae* in disease pathogenesis remains controversial. While its ability to prevent *Clostridium difficile* infection has been documented [44], it has also been proposed as a probiotic to modulate inflammation [45]. In the context of MS, supplementation with the probiotic *S. cerevisiae* has been shown to ameliorate symptoms and the Th1 response in the central nervous system of experimental autoimmune encephalomyelitis (EAE) models. Furthermore, it increases the production of regulatory T cells and enhances microbiota diversity [46]. Other species of the genus *Saccharomyces*, such as *Saccharomyces boulardii*, have been suggested to reduce fatigue, pain and inflammation, promote mental health, and improve the quality of life in pwMS [47].

*Candida* emerges as one of the dominant genera in the mycobiota of the studied population, particularly prominent in pwMS. This genus has significant associations with various diseases and is the fourth most-commonly implicated in CNS infections [48]. Its potential impact on the severity of MS has been previously investigated, revealing that *Candida albicans* isolated from MS patients has elevated levels of proteinase A. This enzyme is known for its role in facilitating penetration and suppressing immune responses, thereby implicating *C. albicans* in the pathogenesis of MS [49]. Furthermore, the interplay between butyrate-producing bacteria and *Candida* has been highlighted [50]. The diminished presence of these bacterial populations in MS may contribute to the modulation of *Candida* colonization in the gut [51]. Additionally, studies have investigated the ability of *C. albicans* and *S. cerevisiae* isolated from MS feces to activate immune cells. Activation assays conducted on the peripheral blood mononuclear cells (PBMCs) from MS patients demonstrated increased activation in response to these fungal species. Activated immune cells have the ability to cross the blood–brain barrier (BBB) and elicit the production of pro-inflammatory cytokines in the brain, potentially exacerbating the inflammatory milieu associated with MS [42].

Furthermore, our investigation revealed disturbances in the balance of fungal–bacterial diversity in MS potentially driven by inter-kingdom interactions. These intricate exchanges between fungi and bacteria, along with the metabolites they produce, exert a significant influence on the proliferation and survival of specific species and can modulate the host immune response [23]. However, our understanding of inter-kingdom connections in commensal microorganisms remains limited. Previous studies focusing on direct bacterial–fungal interactions have depicted fungi as scaffolds for bacteria, facilitating their growth [24,52]. Our results delineate a distinct disease-specific pattern in this inter-kingdom network, characterized by a reduction in the number and intensity of correlations between fungi and bacteria in MS. While many significant fungal–bacterial interactions observed in our healthy cohort are diminished or reversed in the disease state, the MS patient cohort exhibits a greater fungal–bacterial diversity ratio, largely due to increased fungal diversity (partially mitigated by disease-modifying therapies). These alterations may profoundly impact the host immune system and hold significance in diseases such as MS.

Recent findings suggest that the host uses several specific immune mechanisms to eliminate fungi from mucosal surfaces [10]. Among the interactions identified in our study, those involving Proteobacteria and Ascomycota phyla are noteworthy, where the negative correlations observed in the healthy controls are reversed in MS patients. At lower taxonomic levels, interactions include those between *Parasuterellaceae* and *Saccharomyces incertae sedis* and *Meyerozima*, as well as between *Parasutterella* and *Saccharomyces incertae sedis*. Elucidating the significance of these interactions in microbial dysbiosis and the prognosis of MS is imperative.

We do not provide a list of candidates for a potential microbiota-modifying intervention, but highlight the importance of not only specific families of bacteria and fungus, but also the correlation between both kingdoms.

## 5. Conclusions

Our study highlights the dysbiosis of both bacterial and fungal microbiota in MS patients, characterized by significant alterations in biodiversity and composition. The distinct disease-specific pattern of fungal–bacterial interactions suggests that fungi, in addition to bacteria, play a contributory role in the pathogenesis of MS. The complexity of the microbiome ecosystem makes it difficult to design specific microbiota-modifying interventions until we gain a deeper understanding of the relationships among these microorganisms.

## Figures and Tables

**Figure 1 microorganisms-12-00872-f001:**
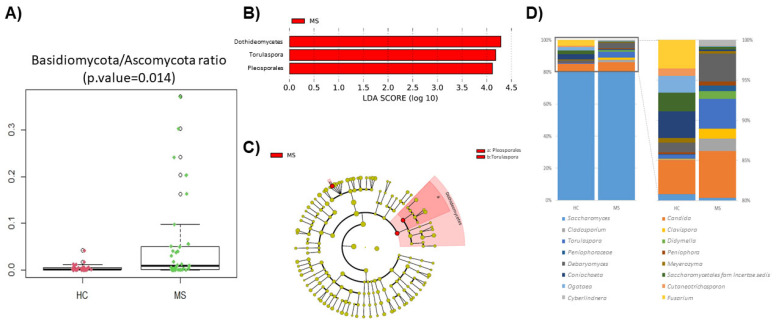
Comparison of the patients’ (MS) and controls’ (HC) mycobiota. (**A**) Box plot showing Basidiomycota/Ascomycota ratio between HC and MS. (**B**) Linear discriminant analysis (LDA) analysis showing differentially abundant bacterial groups as biomarkers determined using the Kruskal–Wallis test (*p* < 0.05) with an LDA score > 2.0. The plotted data represent the microbial differences between in the MS patients (red) and healthy controls. (**C**) Cladogram plotted for the studied population’s mycobiota. Cladograms show the different taxonomic levels by rings; the root of the cladogram denotes the domain fungi, and phyla are represented in the inner ring and genus in the outer one. The plotted data represent the microbial differences between the MS patients (red) and healthy controls. (**D**) Bar plot of the top 10 genera in HC and MS samples, and expansion of the less abundant genera on the right bar plot (note: where resolution at the genus level was not possible, taxa are described at the family level).

**Figure 2 microorganisms-12-00872-f002:**
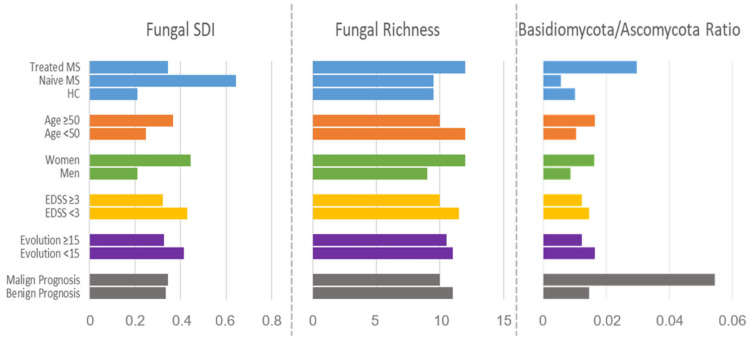
Bar plot of the general parameters of mycobiota composition SDI, richness, and Basidiomyco-ta/Ascomycota ratio, categorized by clinical and demographic parameters. Malign prognosis: EDSS > 3 and less than 15 years of evolution; benign prognosis: EDSS < 3 and more than 15 years of evolution.

**Figure 3 microorganisms-12-00872-f003:**
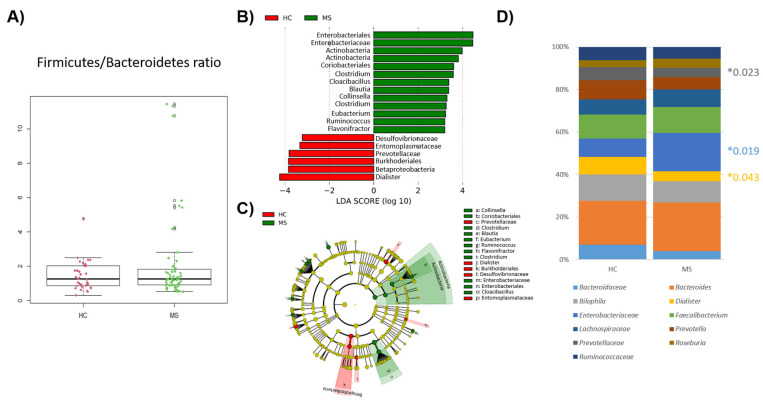
Comparison of patients’ (MS) and controls’ (HC) bacterial microbiota. (**A**) Box plot showing the ratio of Firmicutes to Bacteroidetes between the HCs and MS. (**B**) Linear discriminant analysis (LDA) showing the differentially abundant bacterial groups as biomarkers determined using the Kruskal–Wallis test (*p* <  0.05) with an LDA score > 2.0. The plotted data represent the microbial differences between the MS patients (green) and healthy controls (red). (**C**) Cladogram plotted of the studied population’s microbiota. Cladograms show the different taxonomic levels by rings; the root of the cladogram denotes the domain bacteria, and phyla are represented in the inner ring and genus in the outer one. The plotted data represent the microbial differences between the MS patients (green) and healthy controls (red). (**D**) Bar plot of the top 10 genera in the HC and MS samples (note: where resolution at the genus level was not possible, taxa are described at the family level). Significant differences were indicated with *.

**Figure 4 microorganisms-12-00872-f004:**
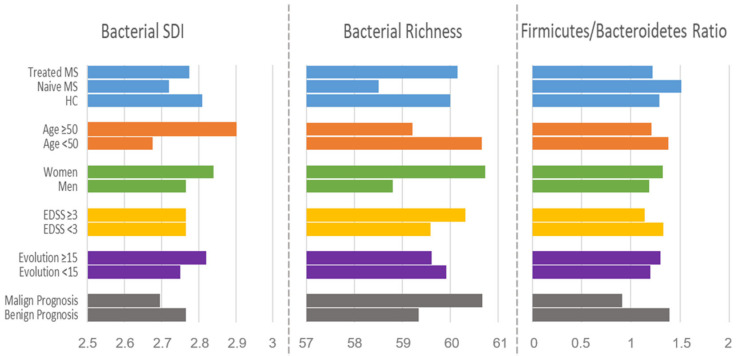
Bar plot of the general parameters of microbiota composition SDI, richness, and Firmicutes/Bacteroidetes ratio, categorized by clinical and demographic parameters. Malign prognosis: EDSS > 3 and less than 15 years of evolution; benign prognosis: EDSS < 3 and more than 15 years of evolution.

**Figure 5 microorganisms-12-00872-f005:**
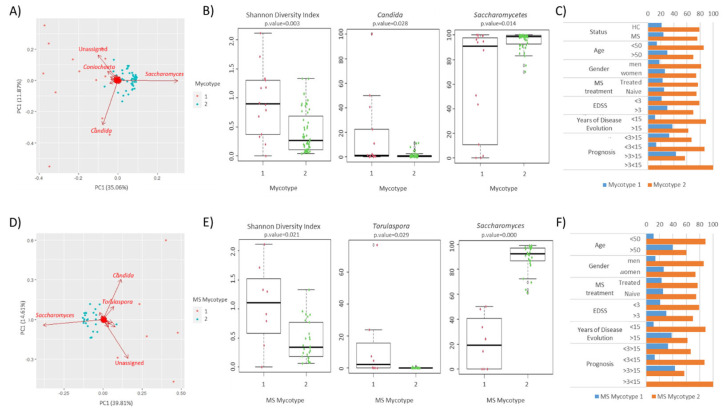
Stratification of the studied population in mycotypes based on their gut mycobiome. (**A**) Scatter plot representing the mycotypes identified among the studied population. Clustered samples are represented in different colors, and the fungal groups that had a higher influence on the samples are represented by arrows. The bidimensional PCoA plot explains 46.9% of the sample’s variability. (**B**) The Shannon diversity index and fungal group abundance box plots showing the main contributors of each mycotype. (**C**) Influence of demographic and clinical factors on the gut fungal profile within each mycotype. (**D**) Scatter plot representing the MS mycotypes identified among the studied population. Clustered samples are represented in different colors, and the fungal groups that had a higher influence on the patient samples are represented by arrows. The bidimensional PCoA plot explains 26.9% of the sample’s variability. (**E**) The Shannon diversity index and fungal genus abundance box plots showing the main contributors to each MS mycotype. (**F**) Influence of demographic and clinical factors on the gut fungal profile within each MS mycotype.

**Figure 6 microorganisms-12-00872-f006:**
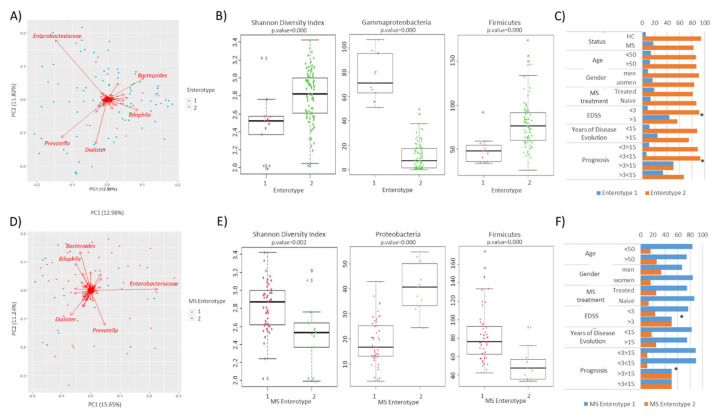
Stratification of the studied population in enterotypes based on their gut microbiome. (**A**) Scatter plot representing the enterotypes identified among the studied population. Clustered samples are represented in different colors, and the bacterial groups that had a higher influence on the samples are represented by arrows. The bidimensional PCoA plot explains 24.8% of the sample’s variability. (**B**) The Shannon diversity index and bacterial group abundance box plots showing the main contributors to each enterotype. (**C**) Influence of demographic and clinical factors on the gut microbial profile within each enterotype (* Fisher’s exact test *p* < 0.05). (**D**) Scatter plot representing the MS enterotypes identified among the studied population. Clustered samples are represented in different colors, and the bacterial groups that had a higher influence on the patient samples are represented by arrows. The bidimensional PcoA plot explains 26.9% of the sample’s variability. (**E**) The Shannon diversity index and bacterial group abundance box plots showing the main contributors to each MS enterotype. (**F**) Influence of demographic and clinical factors on the gut microbial profile within each MS enterotype (* Fisher’s exact test *p* < 0.05).

**Figure 7 microorganisms-12-00872-f007:**
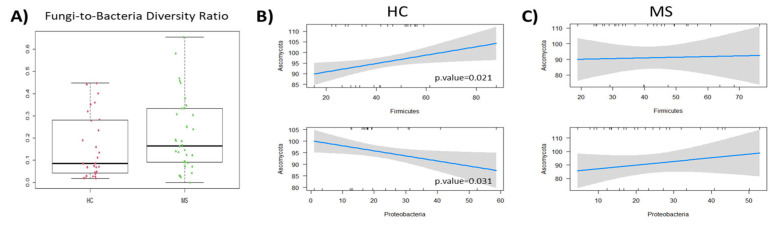
(**A**) Boxplot of the fungi-to-bacteria ratio in the HC and MS patients. (**B**) Correlation of the main fungal phylum Ascomycota and the main bacterial phyla Firmicutes (upper figure) and Proteobacteria (lower figure) in the HCs. (**C**) Correlation of the same dominant fungal and bacterial phyla comparing the HC and the MS patient samples.

**Figure 8 microorganisms-12-00872-f008:**
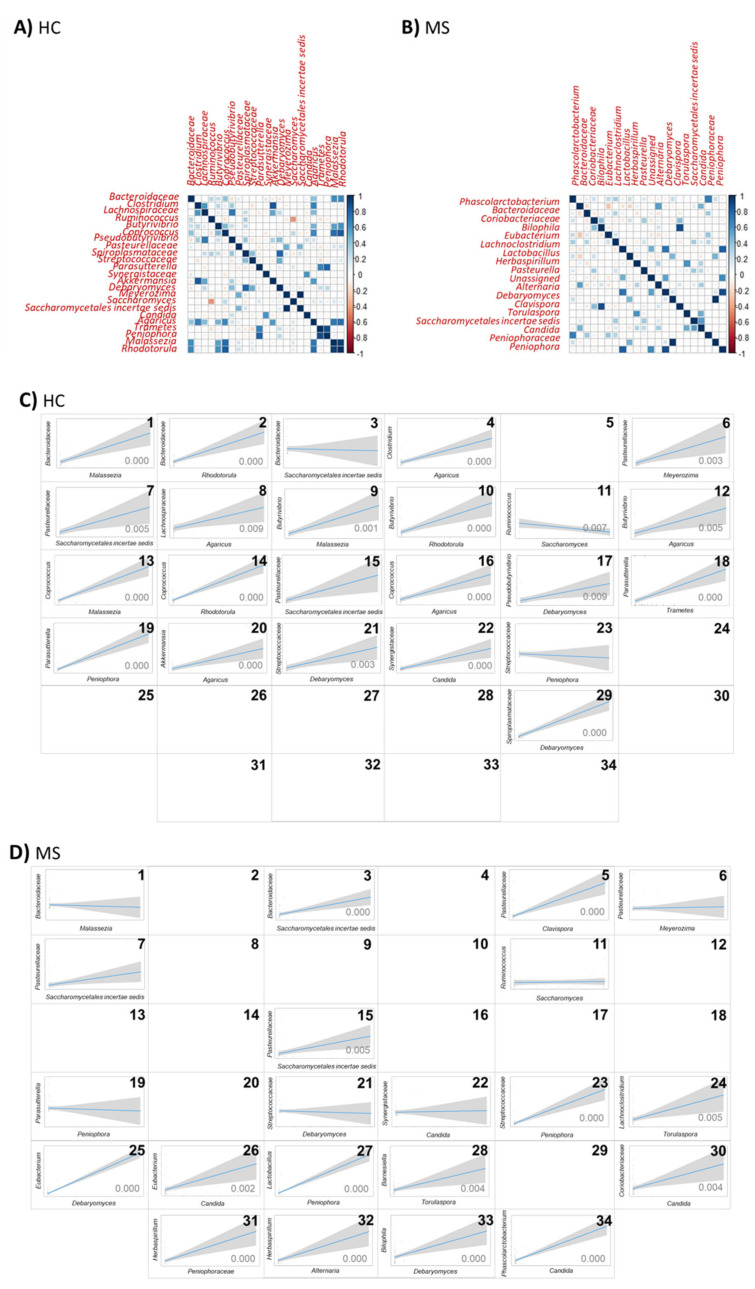
Correlation between the main fungal and bacterial groups (present in at least 10% of the samples and in at least 0.1% on average) in the (**A**) HCs and (**B**) MS. Comparison of the significant correlations between the main fungal and bacterial groups in the (**C**) HCs and (**D**) MS (note: microbial groups not present in 10% of the samples and/or under 0.1% on average were nor represented). **1.** *Bacteroidaceae Malassezia* correlation; **2.** *Bacteroidaceae Rhodotorula* correlation; **3.** *Bacteroidaceae Saccharomycetales incertae sedis* correlation; **4.** *Clostridium Agaricus* correlation; **5.** *Pasteurellaceae Clavispora* correlation; **6.** *Pasteurellaceae Meyerozima* correlation; **7.** *Pasteurellaceae Saccharomycetales incertae sedis* correlation; **8.** *Lachnospiraceae Agaricus* correlation; **9.** *Butyrivibrio Malassezia* correlation; **10.** *Butyrivibrio Rhodotorula* correlation; **11.** *Ruminococcus Saccharomyces* correlation; **12.** *Butyrivibrio Agaricus* correlation; **13.** *Coprococcus Malassezia* correlation; **14.** *Coprococcus Rhodotorula* correlation; **15.** *Parasutterella Saccharomycetales incertae sedis* correlation; **16.** *Coprococcus Agaricus* correlation; **17.** *Pseudobutyrivibrio Debaryomyces* correlation; **18.** *Parasutterella Trametes* correlacion; **19.** *Parasutterella Peniophora* correlation; **20.** *Akkermansia Agaricus* correlation; **21.** *Streptococcaceae Debaryomyces* correlation; **22.** *Synergistaceae Candida* correlation; **23.** *Streptococcaceae Peniophora* correlation; **24.** *Lachnoclostridium Torulaspora* correlation; **25.** *Eubacterium Debaryomyces* correlation; **26.** *Eubacterium Candida* correlation; **27.** *Lactobacillus Peniophora* correlation; **28.** *Barnesiella Torulaspora* correlation; **29.** *Spiroplasmataceae Debaryomyces* correlation; **30.** *Coriobacteriaceae Candida* correlation; **31.** *Herbaspirillum Peniophoraceae* correlation; **32.** *Herbaspirillum Alternaria* correlation; **33.** *Bilophila Debaryomyces* correlation; **34.** *Phascolarctobacterium Candida* correlation.

**Table 1 microorganisms-12-00872-t001:** Clinical and demographic data of the participants in the study.

Variable	Information of Studied Samples n (%)			
**Sample type**	**MS** 62 (63.2%)		**HC** 36 (36.8%)	
**Age**	49.41 (23–89) *		53.47 (35–84) *	
**Sex**	**Female** 43 (69.4%)	**Male** 19 (30.6%)	**Female** 11 (30.5%)	**Male** 25 (69.5%)
**MS type**	**RR** 61 (98.4%)	**SP** 1 (1.6%)	-	
**EDSS**	**≥3** 15 (24.2%)	**<3** 47 (75.8%)	-	
**Evolution (years)**	**≥15** 28 (45.2%)	**<15** 34 (54.8%)	-	
**DMT**	**Treated MS** 48 (77.4%)	**Naive MS** 14 (22.6%)	-	

* Mean (Min-Max); RR: Relapsing-remitting; SP: Secondary progressive; EDSS: Expanded Disability Status Scale; DMT: Disease modifying treatment.

## Data Availability

The datasets presented in this study are being submitted in SRA NCBI repository under the reference SUB14329242.

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
