# Peer review of "Bacteria–Fungi Interactions in Multiple Sclerosis"

_microorganisms, 2024, doi:10.3390/microorganisms12050872_

Round 1
Reviewer 1 Report
Comments and Suggestions for Authors
This paper aimed to profile the gut microbiota and mycobiota of Relapsing–remitting Multiple Sclerosis (RRMS) patients and healthy controls to assess fungal population alterations in multiple sclerosis (MS) and then explore the fungi-bacteria cross-talk in this disease. The authors wanted to gain a broader understanding of the functionality of the gut microbiome in the context of MS. Overall, this manuscript was suggested to be substantially revised and improved. Some major concerns should be carefully addressed as follows.
Major issues:
1) The title of the manuscript was “Cross-Kingdom Microbial Interactions in MS”, which was incorrect! MS was what? What’s the MS? Should I know MS before reading this paper? This title must be firstly revised to “Cross-Kingdom Microbial Interactions in Multiple Sclerosis”.
2) “# Authors contributed equally to the article”, however, only the last author was labeled as the corresponding author. The last two authors were not equal. Now we see this point, they are not of the same equality. How can authors persuade people believe that those two authors contributed equally to the article? Unbelievable.
3) Page 1, line 26, “…with weakned cross-kingdom interactions evident…” What was the “weakned”? So, could the title be changed to “Weakened Cross-Kingdom Microbial Interactions in Multiple Sclerosis” in order to match the abstract?
4) The current title was so big and too simple. The contents were about bacterial-fungal interactions but the title was not detailed. The title seemed to cover a whole wider research area. But the contents were disappointing readers. Could the authors indicate the specific categories of the interactions in the title, to be able to elucidate the relation between the mycobaiota and the MS, or to reveal the crosstalk between fungi and bacteria in this disease.
5) “Ion Torrent sequencing” was not included as a keyword. The keywords were incomplete.
6) Page 1, Line 40, “Relapsing–remitting MS (RR-MS) ”, we saw “RR-MS”; however, in Page 3, Line 101, “RRMS”! Why two were different?
7) The whole introduction should be nicely rewritten.
8) “2.3. DNA extraction and amplification” should be separated into two parts,
2.3. DNA extraction & 2.4. DNA amplification
9) Page 4, Line 176, What was the “16.S”?
10) “2.4. Sample sequencing and data processing” composted too many layers. At least it should be separated into “2.5. Sample sequencing” & “2.6. Data processing”
11) Page 8, Line 316, “3.3. Clustering” was not a subtitle of results. More or less, it looked like “Method”.
12) Page 10, Line 380, “3.4. Pearson correlation” was not a subtitle of results. More or less, it looked like “Method” instead.
13) Was there any graphical abstract (GA) for the manuscript, in correspondence to the Results and Discussions? A good GA was missing.
Comments on the Quality of English Language
Minor or Moderate editing of English language is required
Author Response
This paper aimed to profile the gut microbiota and mycobiota of Relapsing-remitting Multiple Sclerosis (RRMS) patients and healthy controls to assess fungal population alterations in multiple sclerosis (MS) and then explore the fungi-bacteria cross-talk in this disease. The authors wanted to gain a broader understanding of the functionality of the gut microbiome in the context of MS.
Overall this manuscript was suggested t be substantially revised and improved. Some major concerns should be carefully addressed as follows.
Major issues
- The title of the manuscript was “Cross-Kingdom microbial interaction in MS” which was incorrect! MS was what? What´s the MS? Should I know MS before reading this paper? This title must be firstly revised to “Cross-Kingdom microbial interaction in Multiple Sclerosis”
Thanks to the reviewer for noticing this typo, which of course we have corrected in this new version. However, in the uploaded version to the website it appears as Multiple Sclerosis and not as MS. Maybe we use the acronyms in the running title.
- “#Authors contributed equally to the article”, however only the last author was labeled as the corresponding author. The last two authors were not equal. Now we see this point they are not of the same equality. How can authors persuade people believe that those two authors contributed equally to the article? Unbelievable.
Laura Moles and David Otaegui contributed equally to the article, in all the processes of the project, and all the authors agree with this statement. Due to mistakes in uploading the information to the website, the fact that both authors are also the corresponding authors has been mis-transcribed. We have corrected that in this new version of the manuscript.
- Page 1,line 26,”…with weakned cross-kingdom interactions evident…” What was the “weakned”? So, could the title be changed to weakened Cross-Kingdom Microbial Interactions in Multiple Sclerosis in order to match the abstract?
The weakned term refers to the interactions in the MS network compared to the Healthy control.
“Notably, clustering analysis revealed overlapping patterns in bacteriome and mycobiome data for 74% of the participants, with weakned cross-kingdom interactions evident in the altered microbiota of MS patients.”
Therefore, it is a characteristic of the altered microbiota in MS when compared to controls. Besides, and as it has been suggested by the reviewer in another question, we also propose a new title in the new version of the manuscript.
- The current title was so big and too simple. The contents were about bacterial-fungal interactions but the title was not detailed. The title seemed to cover a whole wider research area. But the contents were disappointing readers. Could the authors indicate the specific categories of the interactions in the title, to be able to elucidate the relation between the mycobiota and the MS, or to reveal the crosstalk between fungi and bacteria in this disease.
The title tried to be simple and informative however, it is clear that the reviewer did not agree so we have changed the title to:
Bacteria-Fungi Interactions in Multiple Sclerosis
- “Ion Torrent sequencing” was not included as a keyword. They keywords were incomplete.
Ion Torrent sequencing has been added to the Keywords list.
- Page 1,Line 40, “ Relapsing-Remitting MS (RR-MS)”, we saw “RR-MS”;however, in Page 3,Line 101, “RRMS”! Why two are different?
The nomenclature has been unified through the manuscript to RRMS
- The whole introduction should be nicely rewritten.
We carefully revised and rewritten the introduction to make it more readable, the rest of the manuscript has also been carefully revised.
- “2,2, DNA extraction and amplification should be separated into two parts, 2,3 DNA extraction & 2,4 DNA amplification
The suggested change has been made in the manuscript. Following the next response, we change the title to DNA amplification and sequencing.
- Page 4,Line 176, What was the “16.S”?
The dot in the middle is a typo, thanks for noticing it. Characterization of the bacterial genome has been done by amplifying the 16S ribosomal Gene. We introduce this concept in the DNA amplification section:
“Hypervariable regions V2, V4 and V8, and V3, V6-7 and V9 of the bacterial 16S rRNA gene were amplified”
- “2,4 Sample sequencing and data processing” composted too many layers.At least it should be separated into “2.5 Sample sequencing & “2.6 Data processing”
We agree that the title of this section can be confusing so we changed it to Data Processing since sequencing has been explained in the previous section
“The PCR products obtained from each fecal sample were pooled and sequenced on Ion Torrent PGM equipment (Life Technologies, MA, USA) using a 318 chip.”
“..Between 300 and 400 ng of DNA of each sample was used to perform the amplification and sequencing. Bacterial microbiome study was performed on Ion Torrent PGM equipment (Life Technologies, MA, USA) using a 318 chip.”
- Page 8-Line 316, “3.3 Clustering was not a subtitle of results. More or less,it looked like “Method”.
We understand the reviewer's point of view, however to our understanding this section shows the results and not the methodology. We add a paragraph in methods section explaining the clustering process
- Page 10.Line 380, “3.4 Pearson correlation”was not a subtitle of results. More or less, it looked like “Methos”, instead.
We changed the subtitle to “Pearson correlation results”, since Pearson correlation methodology is already explained in the methods section.
- Was there any graphical abstract(GA) for the manuscript, in correspondence to the results and Discussions? A good GA was missing.
A GA has been added to the manuscript information.
Reviewer 2 Report
Comments and Suggestions for Authors
The research presents an intriguing inquiry into the potential correlation between the microbiome and multiple sclerosis, a well-examined condition where therapies typically offer palliative relief. Regrettably, such findings alone do not suffice to warrant intervention targeting the microbiome and thereby addressing the pathology directly. It is imperative that the authors underscore this aspect in both the discussion and conclusion sections of the study.
Author Response
The research presents an intriguing inquiry into the potential correlation between the microbiome and multiple sclerosis, a well-examined condition where therapies typically offer palliative relief. Regrettably, such findings alone do not suffice to warrant intervention targeting the microbiome and thereby addressing the pathology directly. It is imperative that the authors underscore this aspect in both the discussion and conclusion sections of the study.
We want to thank the reviewer for their comments. The complexity of the microbiome composition is certainly a challenge and despite the efforts of the researchers in the field, we remain far from understanding deeply the composition of this microbiome and therefore designing specific interventions, as the reviewer highlights. We add this idea in the discussion and conclusion as suggested. It is also true that some interventions are being done in Multiple Sclerosis with the intention of modifying the microbiome, however, these interventions only take into account the bacterial kingdom and, it is in this aspect, that we believe that our results will be interesting to the community.
This paragraph has been added to the discussion.
“We do not provide a list of candidates for a potential microbiota-modifying intervention, but highlight the importance of, not only specific families of bacteria and fungus, but also the correlation between both Kingdoms."
The conclusion ends with this sentence in the new version:
"The complexity of the microbiome ecosystem makes it difficult to design specific microbiota-modifying interventions until we gain a deeper understanding of relationships among these microorganisms."
Round 2
Reviewer 1 Report
Comments and Suggestions for Authors
All my concerns and questions have been addressed and the manuscript has been revised properly. Now the manuscript looks just great.
I have no further critical questions on the current manuscript.
To be accepted then.
Reviewer 2 Report
Comments and Suggestions for Authors
The authors have reported the requested changes and can be published